# Considering Predictive Factors in the Diagnosis of Clinically Significant Prostate Cancer in Patients with PI-RADS 3 Lesions

**DOI:** 10.3390/life11121432

**Published:** 2021-12-19

**Authors:** Caleb Natale, Christopher R. Koller, Jacob W. Greenberg, Joshua Pincus, Louis S. Krane

**Affiliations:** 1Department of Urology, Tulane University School of Medicine, New Orleans, LA 70112, USA; cnatale@tulane.edu (C.N.); ckoller@tulane.edu (C.R.K.); jgreenberg@tulane.edu (J.W.G.); jpincus@tulane.edu (J.P.); 2Department of Urology, Southeastern Louisiana Veterans Health Care System, New Orleans, LA 70112, USA

**Keywords:** PI-RADS, prostate cancer, prostate-specific antigen density, MRI, prostate biopsy, targeted biopsy, mpMRI, PSAD, veterans, PSA density

## Abstract

The use of multi-parametric magnetic resonance imaging (mpMRI) in conjunction with the Prostate Imaging Reporting and Data System (PI-RADS) is standard practice in the diagnosis, surveillance, and staging of prostate cancer. The risk associated with lesions graded at a PI-RADS score of 3 is ambiguous. Further characterization of the risk associated with PI-RADS 3 lesions would be useful in guiding further work-up and intervention. This study aims to better characterize the utility of PI-RADS 3 and associated risk factors in detecting clinically significant prostate cancer. From a prospectively maintained IRB-approved dataset of all veterans undergoing mpMRI fusion biopsy at the Southeastern Louisiana Veterans Healthcare System, we identified a cohort of 230 PI-RADS 3 lesions from a dataset of 283 consecutive UroNav-guided biopsies in 263 patients from October 2017 to July 2020. Clinically significant prostate cancer (Gleason Grade ≥ 2) was detected in 18 of the biopsied PI-RADS 3 lesions, representing 7.8% of the overall sample. Based on binomial analysis, PSA densities of 0.15 or greater were predictive of clinically significant disease, as was PSA. The location of the lesion within the prostate was not shown to be a statistically significant predictor of prostate cancer overall (*p* = 0.87), or of clinically significant disease (*p* = 0.16). The majority of PI-RADS 3 lesions do not represent clinically significant disease; therefore, it is possible to reduce morbidity through biopsy. PSA density is a potential adjunctive factor in deciding which patients with PI-RADS 3 lesions require biopsy. Furthermore, while the risk of prostate cancer for African-American men has been debated in the literature, our findings indicate that race is not predictive of identifying prostate cancer, with comparable Gleason grade distributions on histology between races.

## 1. Introduction

The use of multiparametric magnetic resonance imaging (mpMRI) in the diagnosis of prostate cancer has increased over the past decade, due to the potential of this technology to increase detection rates of clinically significant prostate cancer and minimize overdiagnosis of low-risk disease [1,2]. The use of mpMRI prior to biopsy may also lead to a reduction in overdiagnosis of clinically insignificant lesions and reduce unnecessary biopsies [3]. New evidence suggests that the use of mpMRI may also serve to guide which patients may be appropriate for active surveillance protocols [4]. The Prostate Imaging Reporting and Data System (PI-RADS) was developed to standardize the imaging acquisition and reporting of prostate mpMRI findings in order to aid in providing clinicians with the ability to make treatment decisions and guidelines. PI-RADS v2 was developed to further simplify and standardize the acquisition, interpretation, and reporting of prostate mpMRI exams, with the intention that the system should evolve according to clinical consensus [5,6].

Clinical guidance using the PI-RADS v2 system is limited when considering lesions of score 3, which are noted to be equivocal for clinically significant prostate cancer. While the positive predictive value of clinically significant prostate cancer is often quite low for PI-RADS 3 lesions—just 13% based on a recent meta-analysis [7]—inter-reader reliability remains a concern [8]. There is significant variability between reported prevalence and clinically significant prostate cancer detection among studies in the existing literature [9]. Biopsy is a reasonable choice if there is continued suspicion of clinically significant disease. Close surveillance is also a reasonable option in patients who would like to avoid biopsy and who are willing and able to adhere to strict monitoring protocols [10]. Although further elucidation of clinically predictive factors of clinically significant prostate cancer is needed in order to better inform clinical decision making for patients with identified PI-RADS 3 lesions, there are insufficient trial data to define clinical parameters useful in this decision-making process [9]. The aim of this study was to evaluate the association between clinical characteristics and clinically significant prostate cancer in patients whose mpMRI images contained lesions determined to be PI-RADS 3.

## 2. Materials and Methods

We analyzed mpMRI data from a prospectively maintained database of all patients undergoing mpMRI fusion biopsy at a single veterans’ healthcare facility from October 2017 to July 2020. Patients whose MRI examinations revealed PI-RADS score 3 lesions were included in this study. Clinical and radiological characteristics were recorded in the database and analyzed as part of this study; these included age, race, PSA, PSA density, MRI prostate volume, BMI, active surveillance status, and anatomic lesion location.

MRI examinations were carried out according to standard protocols as defined by PI-RADS v2 and standard clinical practice [11]. The study was conducted in accordance with the guidelines of the Declaration of Helsinki, and approved by the Institutional Review Board of Southern Louisiana Veterans Healthcare System (#563–629). The MRI consisted of T2-weighted images (T2W) and diffusion-weighted images (DWI) taken on a 3T system. Contrast enhancement was performed. mpMRI interpretation was conducted by radiologists experienced in prostate MRI reading and familiar with PI-RADs scoring. PI-RADS 3 was defined as per the PI-RADS v2.1 standard.

Prostate biopsies were performed via the transrectal approach. Both template and targeted biopsies were obtained for each patient. MRI-targeted biopsies utilized the UroNav MRI/ultrasound-guided fusion biopsy system (Koninklijke Philips, Amsterdam, Netherlands). Histopathology was reported by fellowship-trained genitourinary pathologists. If the pathologist had concerns about their reporting, a report by an additional pathologist on staff was obtained. If further concerns remained, histopathology specimens were sent for external central pathological review at the Joint Pathology Center. Clinically significant prostate cancer was defined as Gleason Grade Group ≥2 (Gleason score 7 or greater).

Data were analyzed using JMP Statistical Discover 14.1 (SAS Institute Inc., Cary, NC, USA). Binomial regression analysis was performed to characterize the associations between recorded clinical or radiological characteristics and clinically significant prostate cancer. Odds ratios and confidence intervals were calculated. Pearson’s chi-squared test was used to evaluate statistical significance between lesion location and clinically significant prostate cancer. Stacked bar plots were generated using R computational language version 4.1.2. Fisher’s exact test was employed to evaluate racial differences in prostate cancer detection. All statistical significance was defined as *p* < 0.05, and tests were two-sided.

## 3. Results

### 3.1. Study Cohort

We analyzed a sample of 263 consecutive veteran patients who received mpMRI followed by UroNav-guided biopsies at our institution. mpMRI identified 546 lesions including 0 PI-RADS 1, 12 PI-RADS 2, 214 PI-RADS 4, and 90 PI-RADS 5 lesions. We completed 283 MRI fusion biopsies in 263 patients. There were 230 PI-RADS 3 lesions identified in 144 patients. These 230 PI-RADS 3 lesions were included in the study cohort. The mean age of patients in this cohort was 68.2 (63.9–70.8) years, and the median PSA was 6.26 (4.66–9.12) ng/mL. Median prostate volume measured by MRI was 53.6 (38.2–75.2) mL. Characteristics of the study cohort are presented in Table 1.

### 3.2. Biopsy Results

Prostate cancer was detected in 31 (13.5%) of 230 biopsied PI-RADS 3 lesions. Of these, 18 constituted clinically significant prostate cancer (Gleason Grade ≥ 2), representing 7.8% of the overall sample. Histopathology found 11 of these biopsy samples to contain Gleason Grade 2 disease, 6 biopsy samples to contain Gleason Grade 3 disease, and 1 biopsy sample to contain Gleason Grade 4 disease.

A total of 10 out of 75 (13.3%) PI-RADS 3 lesions with PSA density ≥0.15 ng/mL/mL contained clinically significant prostate cancer, compared to 8 out of 155 (5.2%) PI-RADS 3 lesions with PSA density <0.15 ng/mL^2^.

There were 67 biopsies that contained exclusively PI-RADS 3 lesions without additional higher scored PI-RADS lesions within the prostate. Considering this subset of biopsies, there were 14 instances of clinically significant prostate cancer detected in the template biopsies without clinically significant disease within the region-of-interest biopsies. The majority of these (9 out of 14, 64.3%) were in patients with PSA density ≥0.15 ng/mL/mL, and most (10 out of 14, 71.4%) were detected in regions distinct from the region of interest (contralateral lobe of the prostate).

Patients were then subdivided into groups with one PI-RADS 3 lesion or more than one PI-RADS 3 lesion. Seventy-eight patients were found to have one PI-RADS 3 lesion, while sixty-six had more than one PI-RADS 3 lesion (Table 2). This was found to be a total of 78 patients with one PI-RADS 3 lesion, 49 with two, 14 with three, and 3 subjects were found to have four lesions on mpMRI. Demographics, co-morbidities, PSA at the time of biopsy, and MRI prostate volume were all found to be statistically comparable between groups. When considering clinically significant prostate cancer detected by targeted biopsy, an increasing number of PI-RADS 3 lesions trended toward being protective against GGG ≥ 1. This was not statistically significant (*p* = 0.015).

### 3.3. Racial Compassions of Disease 

The majority of patients enrolled into this study identified themselves as being of African-American (AA) race or heritage. With small cohorts of AA men reported in the literature, we felt it necessary to compare patient demographics, prostate features, and biopsy findings between racial groups. Patients were split into two groups: AA and non-AA. Our non-AA subcohort contained 94% (*n* = 31) Caucasian (CA) and 6% (*n* = 2) men self-reporting a race other than AA or CA. Non-AA and AA men had a median age of 68.15 and 68.2 years, respectively (*p* = 0.65). Additionally, patient BMI, PSA, MRI prostate volume, and PSA density at the time of biopsy were comparable between racial groups, with *p*-values of 0.47, 0.42, 0.25, 0.67, and 0.88, respectively. Next, the anatomical locations of regions of interest and lesions positive for prostate cancer were evaluated between racial groups. The distribution of regions of interest throughout the prostate was comparable between races (*p* > 0.05). Additionally, the anatomical locations of regions of interest positive for prostate cancer and clinically significant prostate cancer were similar between races.

From the 230 PI-RADS 3 lesions, 55 were identified in non-AA and 175 in AA men. Prostate cancer was detected in 5 (9%) non-AA and 26 (14.8%) AA patients. A Fisher’s exact test comparing the numbers of positive and negative biopsies between races yielded a *p*-value of 0.37. Of the biopsies positive for prostate cancer, the rates of clinically significant prostate cancer for non-AA and AA men were 60% and 58%, respectively. The distribution of Gleason Grade between races can be found in Figure 1. For both the non-AA and AA patient cohorts, pathology results from the majority of biopsies recorded benign findings. For AA men, 11 subjects were diagnosed with Gleason Grade Group 1 disease, 9 were diagnosed with Gleason Grade Group 2, 5 with Gleason Grade Group 3, and 1 with Gleason Grade Group 4 upon histological evaluation. For non-AA men, two patients were diagnosed with Gleason Grade Group 1, two with Gleason Grade Group 2, and one with Gleason Grade Group 3 upon histological evaluation.

### 3.4. Predicting Clinically Significant Disease 

Clinically significant prostate cancer was most prevalent in lesions located within the transitional zone (16.1%), followed by lesions within the anterior lobe (14.6%) and lesions located within the peripheral zone (7.0%). The location of the lesion was not shown to be a statistically significant predictor of prostate cancer overall (*p* = 0.87), nor of clinically significant disease (*p* = 0.16). Based on univariate regression analysis, prostate volume was significantly different (*p* = 0.011) in biopsies that contained clinically significant prostate cancer compared to those that did not. PSA density showed a trend toward difference between the two groups, but did not meet statistical significance (*p* = 0.061).

Based on multivariate regression analysis of patient characteristics, PSA density and PSA were shown to be significant predictors of clinically significant disease (Table 3). PSA densities of 0.15 or greater portended clinically significant prostate cancer, when compared to PSA densities of less than 0.15 (odds ratio: 5.39; 95% CI: 1.53–18.98; *p* = 0.009). Age, BMI, and active surveillance protocol status were not statistically significant predictors of clinically significant prostate cancer. Furthermore, race was not found to be a predictor of clinically significant prostate cancer, with an odds ratio of 1.562 (95% CI: 0.415–5.88; *p* = 0.64). Prostate volume was not included in the multivariate analysis, and was used to calculate PSA density. An ROC curve was generated to quantify the overall accuracy of PSA density in predicting clinically significant prostate cancer detection within PIRADS 3 lesions (Figure 2). The optimal area under the curve (AUC) was calculated to be 64.2%.

## 4. Discussion

The classification of a lesion as PI-RADS 3 indicates that the presence or absence of clinically significant prostate cancer is equivocal in a lesion of interest. PI-RADS was developed to standardize the acquisition, interpretation, and reporting of prostate MRI findings. PI-RADS v2 aimed to simplify the classification of lesions necessitating biopsy. The classification of PI-RADS v2 score 3 lesions is a notable deficiency in this system, as the ambiguity of this category does not offer precise guidance on how to proceed clinically [12,13]; this may introduce variation in how to proceed. Some physicians may choose to biopsy all PI-RADS 3 lesions, while others may consider other aspects of the clinical picture in order to decide whether biopsy may be appropriate. Optimal training in mpMRI interpretation may help to mitigate this variation. At present, an optimal standardized training protocol in PI-RADS v2 has not been developed, and the PI-RADS system is likely to continue to evolve as collective experience informs meaningful adjustment.

Variation also exists in the proportion of lesions reported to contain clinically significant disease upon biopsy. In this study, we report that 7.8% of biopsied PI-RADS 3 lesions were found to contain clinically significant prostate cancer. Other reports in the literature describe rates of 4.4% [14], 4.7% [15], 22.9% [16], 14.6% [17], and 22% [18]. It should be noted that only one of these studies contained AA patients, reporting only six patients with PI-RADS 3 lesions. This discrepancy may be partially attributed to a noted learning curve in prostate MRI interpretation, with accuracy improving over time [19]. Unique protocols and radiologists’ preferences at different institutions may also modify which lesions are assigned an equivocal risk based on protocols concerning which PI-RADS 3 lesions should be biopsied [10].

For these reasons, emphasis has been placed on further developing adjuncts to the mpMRI in order to determine which PI-RADS 3 lesions merit biopsy. A stated goal of this study was to investigate which clinical characteristics, if any, could represent indicators of clinically significant disease. In this study, we demonstrated a statistically significant tendency towards clinically significant prostate cancer in those individuals whose PSA density was ≥0.15 ng/mL^2^, who were more than 2.5 times more likely to show significant disease on histopathology compared to individuals with PSA density <0.15 ng/mL^2^.

These results demonstrating the value of PSA density as an adjunct to mpMRI in PI-RADS 3 lesions are concordant with existing literature. In this series, if patients with PI-RADS 3 lesions with PSA density <0.15 ng/mL^2^ had been clinically monitored, then 55.7% (128/230) would have been spared biopsy, while 3.5% (8/230) would have experienced delayed or missed diagnosis of clinically significant prostate cancer. We also illustrated that, based on univariate analysis, smaller prostate volume was associated with propensity for prostate cancer, as has been reported previously [16]. Washino et al. retrospectively reviewed 288 patients, including 43 with PI-RADS 3 lesions [20]; they found that a PI-RADS v2 score of 3 and a PSA density of ≥0.15 ng/mL^2^ yielded clinically significant prostate cancer in 54% of lesions, compared to no clinically significant prostate cancer in the 6 lesions with a PI-RADS v2 score of 3 and a PSA density of <0.15 ng/mL^2^. Görtz et al. demonstrated that PSA density was a significant predictor of clinically significant prostate cancer among a sample of 101 PI-RADS 3 lesions [21]; furthermore, the authors concluded that forgoing biopsy in patients demonstrating PSA density <0.1 ng/mL^2^ would have resulted in a 43% reduction (43/101) in biopsies in PI-RADS 3 patients and resulted in missing 2% (1/43) of clinically significant prostate cancer cases.

Secondary biomarkers and nomograms, in addition to routine clinical markers in the treatment of prostate cancer, may represent additional adjunctive criteria to a PI-RADS score of 3 for informing subsequent clinical steps [10]. The size of the index lesion in PI-RADS 3 lesions may represent a future adjunctive factor for consideration. Lesions found to be 1.5 cm or greater in diameter would result in an upgrade from a Likert score of 4 to 5 according to the PI-RADS v2 protocol. Stanley et al. reported that PI-RADS 3 lesions sized less than 0.5 cm were not likely to represent clinically significant disease [22]. It has been suggested that patients with PI-RADS 3 lesions be clinically surveilled, while those with lesions measuring 0.5 mL or greater be treated with targeted biopsy [13]. In a retrospective study of 99 patients, Luis et al. reported improved sensitivity and specificity for clinically significant prostate cancer when considering lesions greater than 0.5 mL and PSA density ≥0.15 ng/mL^2^ [23], lending evidence to this proposed practice.

While this study determined that PSA density was a significant predictor of malignancy, race was not statistically significant. In 2016, DeSantis et al. analyzed prostate cancer incidence and mortality by race from the SEER database [24]. These investigators found that African-American men had increased prostate cancer incidence in both locally advanced and metastatic disease. When evaluating disease-specific mortality, African-Americans were over two times more likely to die from prostate cancer compared to non-Hispanic Caucasian men. mpMRI has been showed to improve early detection of clinically significant prostate cancer. While mpMRI has improved cancer detection, racial disparities still exist among those undergoing mpMRI-targeted biopsies. Patel et al. analyzed a group containing 53 African-American men with PI-RADS 3 lesions [25]; amongst all PI-RADS lesions, African-Americans were 1.64 times more likely to be diagnosed with prostate cancer over their cohort of non-AA patients. While this study gives clinicians valuable insight into anterior lesions, increased prostate cancer risk was identified among all biopsies, and a delineated PI-RADS score analysis (specifically PI-RADS 3) between racial groups was not reported. Conversely, Walton et al. did not find increased prostate cancer risk for African-American men [26]; however, Walton et al.’s study was limited by small sample size (31 AA men, 27 with PI-RADS 3) and no PI-RADS delineated analysis between races. Our findings and others published from our lab show race to not be predictive of identifying prostate cancer or clinically significant prostate cancer [27,28,29].

While the studies supporting increased prostate cancer risk in African-American men have been debated, few consider the effect of access to healthcare on their results. The vast majority of published literature on African-American men and prostate cancer has been conducted on patients with varying levels of insurance and access to healthcare. The data represented in this study were collected from a single-payer healthcare system, where access to healthcare was constant across racial groups. A total of 99 African-American men were analyzed, and their PI-RADS 3 targeted biopsy results were compared to those of non-AA counterparts. First, racial groups were found to have comparable demographics in terms of age, BMI, PSA, MRI prostate volume, and PSA density at the time of their biopsy. Upon anatomical and histological evaluation, comparable region of interest locations, rates of prostate cancer, and Gleason Grade distribution were identified between AA and non-AA men. Coughlin et al. highlighted that that African-American men experience significant socioeconomic pressures in the United States [30]. These stressors can manifest as inconstant office visits and loss of follow-up due to factors such as the costs associated with mpMRI imaging and urological consultation. When these stressors are minimized, our data showed rates and severity of prostate cancer among African-American men with PI-RADS 3 lesions identified on mpMRI to be comparable to those of the general population.

The limitations of this study should be considered. While the PI-RADS system was developed with a methodology for standardized classification, there is inherent inter-observer variability between radiologists. We conducted this study within a single center, but slight variations in practice between centers may serve to lessen generalizability. The proportion of lesions detected with clinically significant disease was low. While this was an important finding, it may have resulted in inadequate power to detect statistically significant risk factors for clinically significant prostate cancer. Furthermore, certain factors could not be assessed due to incomplete data. In some cases, patients were unwilling or unable to provide this data for use in the study of prospectively collected results.

## 5. Conclusions

In this single-center study, we have demonstrated that the majority of PI-RADS 3 lesions do not represent clinically significant disease and, therefore, the opportunity to reduce morbidity through biopsy. PSA density is a potential adjunctive factor in deciding which patients with PI-RADS 3 lesions require biopsy. Additionally, the current literature debates increased prostate cancer risk among men of African-American heritage. However, this study found similar prostate cancer and clinically significant prostate cancer detection rates when African-American men were compared to their non-AA counterparts.

## Figures and Tables

**Figure 1 life-11-01432-f001:**
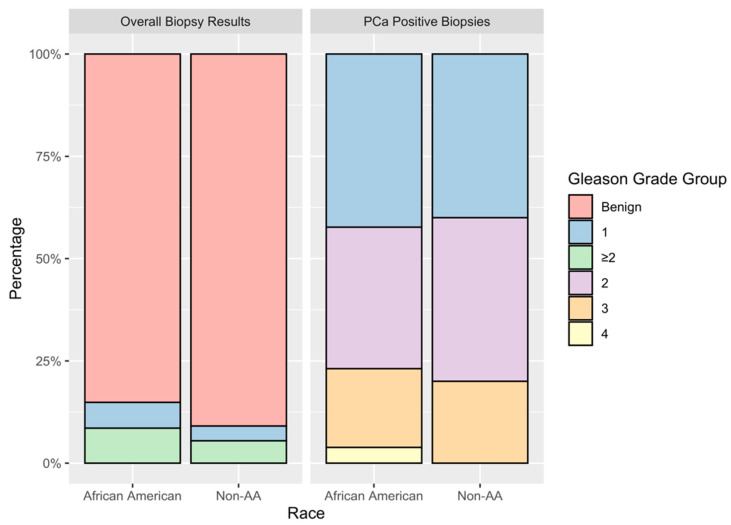
Stacked bar graph representing the Gleason Grade Group distribution of pathological results of PI-RADS 3 lesions. Racial groups showed comparable distributions of prostate cancer overall, and of clinically significant disease.

**Figure 2 life-11-01432-f002:**
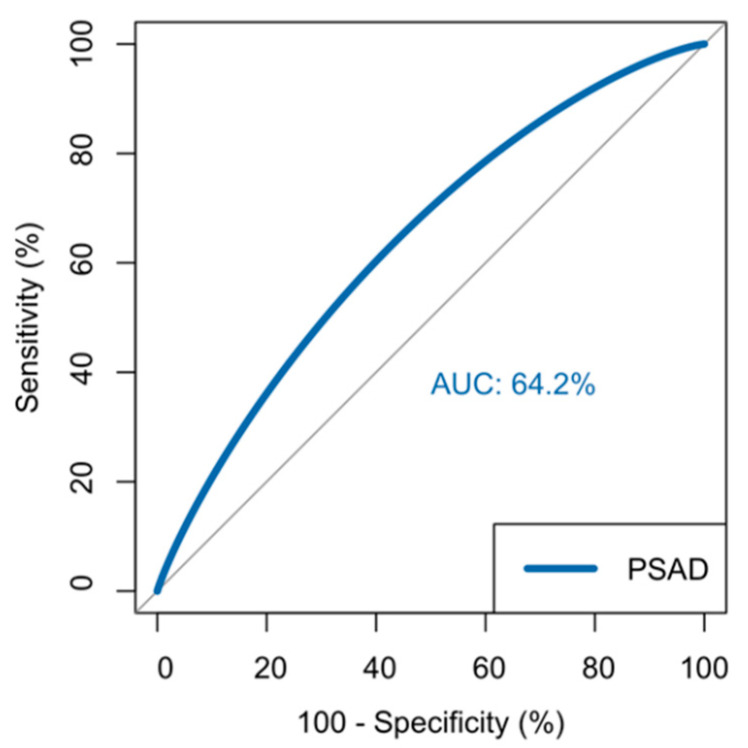
Smoothed ROC curve demonstrating the accuracy of PSA density in predicting clinically significant prostate cancer.

**Table 1 life-11-01432-t001:** Demographics of 144 patients with 230 MRI PI-RADS 3 lesions.

Variable	Total PI-RADS 3 Lesions
Median Age (IQR)	68.2 (63.9–70.8) years
Median PSA (IQR)	6.26 (4.66–9.12) ng/mL
Median PSA Density (IQR)	0.13 (0.08–0.22) ng/mL^2^
Median MRI Prostate Volume (IQR)	53.6 (38.2–75.2) mL
Median BMI (IQR)	28.9 (25.7–32.4)
*n*, On Active Surveillance (%)	94 (41%)
**Race (%)**	––––––
*n*, African-American	99 (75%)
*n*, Caucasian	31 (23%)
*n*, Other	2 (1.5%)
**Location (%)**	––––––
*n*, Peripheral Zone	158 (69%)
*n*, Transitional Zone	31 (13%)
*n*, Anterior Lesion	41 (18%)

**Table 2 life-11-01432-t002:** Rates of identifying Gleason Grade Group ≥1 by mpMRI-targeted biopsy.

Number of PIRADS 3 Lesions	Rates of GGG ≥1 Cancer	*p*-Value
1 (*n* = 78)	18%	–––
2 (*n* = 98)	12.2%	–––
3 (*n* = 42)	11.9%	–––
4 (*n* = 12)	0%	0.15

**Table 3 life-11-01432-t003:** Odds ratios of clinically significant prostate cancer for selected characteristics.

Variable	Odds Ratio	95% CI	*p*-Value
Age	1.032	0.942–1.130	0.493
Race	1.562	0.415–5.88	0.641
BMI	0.970	0.888–1.060	0.495
PSA	0.835 *	0.687–1.016	0.031 *
PSA Density	5.39 *	1.531–18.980	0.009 **
On Active Surveillance	1.027	0.369–2.858	0.96

* *p* < 0.05; ** *p* < 0.01.

## Data Availability

Data are available on request due to privacy restrictions. The data presented in this study are available upon request from the corresponding author. The data are not publicly available due to concerns about overexposing protected health information.

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
