# Peer review of "Considering Predictive Factors in the Diagnosis of Clinically Significant Prostate Cancer in Patients with PI-RADS 3 Lesions"

_life, 2021, doi:10.3390/life11121432_

Round 1

Reviewer 1 Report

I consider the article well designed and presented.
my suggestions are:
in the results adequately describe the tables and figures. I suggest putting the figure or table immediately after its description

Author Response

We would like to say thank you to the editors and reviews who took the time to review our manuscript. Below you can find our responses to the review’s comments in bold.

my suggestions are:
in the results adequately describe the tables and figures. I suggest putting the figure or table immediately after its description

Thank you for the suggestion, we have integrated the tables and figures into the results section and have ensured that each is described within the body.

Reviewer 2 Report

The manuscript aims to analyze the association between clinical characteristics and clinically significant PCa in patients with PI-RADS 3 lesions at mpMRI.

First of all, the abstract should be improved as the aim of the study is not clearly reported.

The paragraph "Biopsy Results" is quite confusing, as in some cases the total referred to is implicitly 230, in other cases 283... Please modify this paragraph and explicit these numbers.

In the "Discussion" section you should mention the limitation of your study.

Regarding the references, in line 35 (first page) there is ref. [30], which is "impossible", as it is written at the beginning. Furthermore, Ref. [28] is reported after ref. [29] (line 279).

Finally, an extensive English revision is required.

Author Response

We would like to say thank you to the editors and reviews who took the time to review our manuscript. Below you can find our responses to the review’s comments in bold.

The manuscript aims to analyze the association between clinical characteristics and clinically significant PCa in patients with PI-RADS 3 lesions at mpMRI.

First of all, the abstract should be improved as the aim of the study is not clearly reported.

We have modified the abstract including by adding This study aims to better characterize the utility of PI-RADS 3 and associated risk factors in detecting clinically significant prostate cancer” to clearly state the objective of the study.

The paragraph "Biopsy Results" is quite confusing, as in some cases the total referred to is implicitly 230, in other cases 283... Please modify this paragraph and explicit these numbers.

We have revised this section for clarity. While the 230 refers to PIRADS 3 lesions that were biopsied, the 283 represents the number of consecutive MRI fusion biopsies that took place at our institution from the start of this study until our data-lock.  We have changed the wording to better emphasize this point and have made additional changes throughout the paragraph.

In the "Discussion" section you should mention the limitation of your study.

A section on limitations has been added to the Discussion section of the study.

Regarding the references, in line 35 (first page) there is ref. [30], which is "impossible", as it is written at the beginning. Furthermore, Ref. [28] is reported after ref. [29] (line 279).

The order of the references has been modified to conform to ascending order.

Finally, an extensive English revision is required.

This manuscript was reviewed and edited for grammar, punctuation and syntax as noted by highlighter within the manuscript.

Round 2

Reviewer 2 Report

Now the manuscript has been greatly improved and can be considered for publication.